# Closure of the Eastern Paleo-Asian Ocean: Constraints from the Age and Geochemistry of Early Permian Zhaojinggou Pluton in Inner Mongolia (North China)

**Guang-Yao Li** [1,2,3], **Liang Qiu** [3,*], **Zhi-Dan Li** [1,2,*], **Lei Gao** [3], **Chao Fu** [1,2], **Jia-Ying Wang** [1,2], **Qi Zhang** [1,2], **Jia-Run Tu** [1,2] and **Teng-Fei Ge** [4]

1 Tianjin Center, China Geological Survey, Tianjin 300170, China; lguangyao@mail.cgs.gov.cn (G.-Y.L.); fchao@mail.cgs.gov.cn (C.F.); wjiaying@mail.cgs.gov.cn (J.-Y.W.); zqi@mail.cgs.gov.cn (Q.Z.); tjiarun@mail.cgs.gov.cn (J.-R.T.)
2 North China Center for Geoscience Innovation, China Geological Survey, Tianjin 300170, China
3 State Key Laboratory of Geological Processes and Mineral Resources, School of Earth Sciences and Resources, China University of Geosciences, Beijing 100083, China; leigao@cugb.edu.cn
4 China Aero Geophysical Survey and Remote Sensing Center for Natural Resources, Beijing 100083, China; gtengfei@mail.cgs.gov.cn
* Correspondence: qiul@cugb.edu.cn (L.Q.); lzhidan@mail.cgs.gov.cn (Z.-D.L.)

**Abstract:** The closing time of the Paleo-Asian Ocean and the tectonic evolution of the northern margin of the North China Craton are still controversial. The geochronology and geochemistry of the Zhaojinggou monzogranite pluton provide new constraints on the late Paleozoic tectonic evolution and the closure time of the Paleo-Asian Ocean in the southern Central Asian Orogenic Belt (CAOB). The monzogranite yielded a zircon U-Pb age of $286.7 \pm 1.2$ Ma. Due to the characteristics of low–moderate $Mg^\#$ values (25.87–39.21), low $Fe_2O_3^T$ values (1.13–1.72), and A/CNK > 1, we show that the pluton is weak peraluminous, high in potassium calc–alkaline series, and displays the feature of S-type granite. The total REE content is low, the distribution curve is right dipping, and the LREE is enriched; the δEu average value is 1.32 (1.11–1.54). The granite presents relatively high $(^{87}Sr/^{86}Sr)_i$ values of 0.712345–0.713723, low $\varepsilon_{Nd}(t)$ values of $-8.89$–$-8.21$ (an average value of $-8.56$), and a $T_{DM2}$ of 1718–1773 Ma. Furthermore, the zircon in situ Hf isotopic analysis shows $^{176}Hf/^{177}Hf$ ratios of 0.282342 to 0.282614, low $\varepsilon_{Hf}(t)$ values of $-9.27$–0.38 (mean $-4.74$), and a $T_{DM2}$ of 1275–1887 Ma. Additionally, high field strength elements such as Nb, Ta, and Ti are depleted, and large ion lithophile elements, e.g., Rb, Ba, K, and Sr, are enriched. The above features of the Zhaojinggou monzogranite indicate that the pluton was derived from late Paleoproterozoic to Mesoproterozoic lower crustal mafic materials. By discussing the genesis and tectonic implications of the pluton massif, we propose that the Zhaojinggou monzogranite represents a magmatic event caused by the crustal–mantle interaction during the southward subduction of the eastern Paleo-Asian Ocean in the northern margin of the North China Craton during the Early Permian.

**Keywords:** S-type granite; late Paleozoic; Paleo-Asian Ocean; Zhaojinggou monzogranite pluton; Central Asian Orogenic Belt





## 1. Introduction

The closure time of the eastern Paleo-Asian Ocean in the northern margin of the North China Craton (NCC) is still controversial [1]. A paleomagnetic study showed that the Paleo-Asian Ocean closed before the Early Permian [2]. However, some authors have indicated that the ocean finally closed in the Late Permian to early Middle Triassic [3], and others propose that the final closure time lasted until the middle Late Triassic [4]. Most scholars believe that the Suolun ophiolite belt was the suture zone of the collision between the North China plate (NCC) and the Siberia plate, and its formation was closely

related to the closure of the Paleo-Asian Ocean [1,5–8]. The discovery of Zhaojinggou weak peraluminous S-type granite in the south of the Suolun suture zone is evidence of the existence of syn-collisional granite in the study area (Figure 1a,b), which may provide an opportunity to reconstruct the orogenic process, crustal evolution, and tectonic history of the eastern Paleo-Asian Ocean.

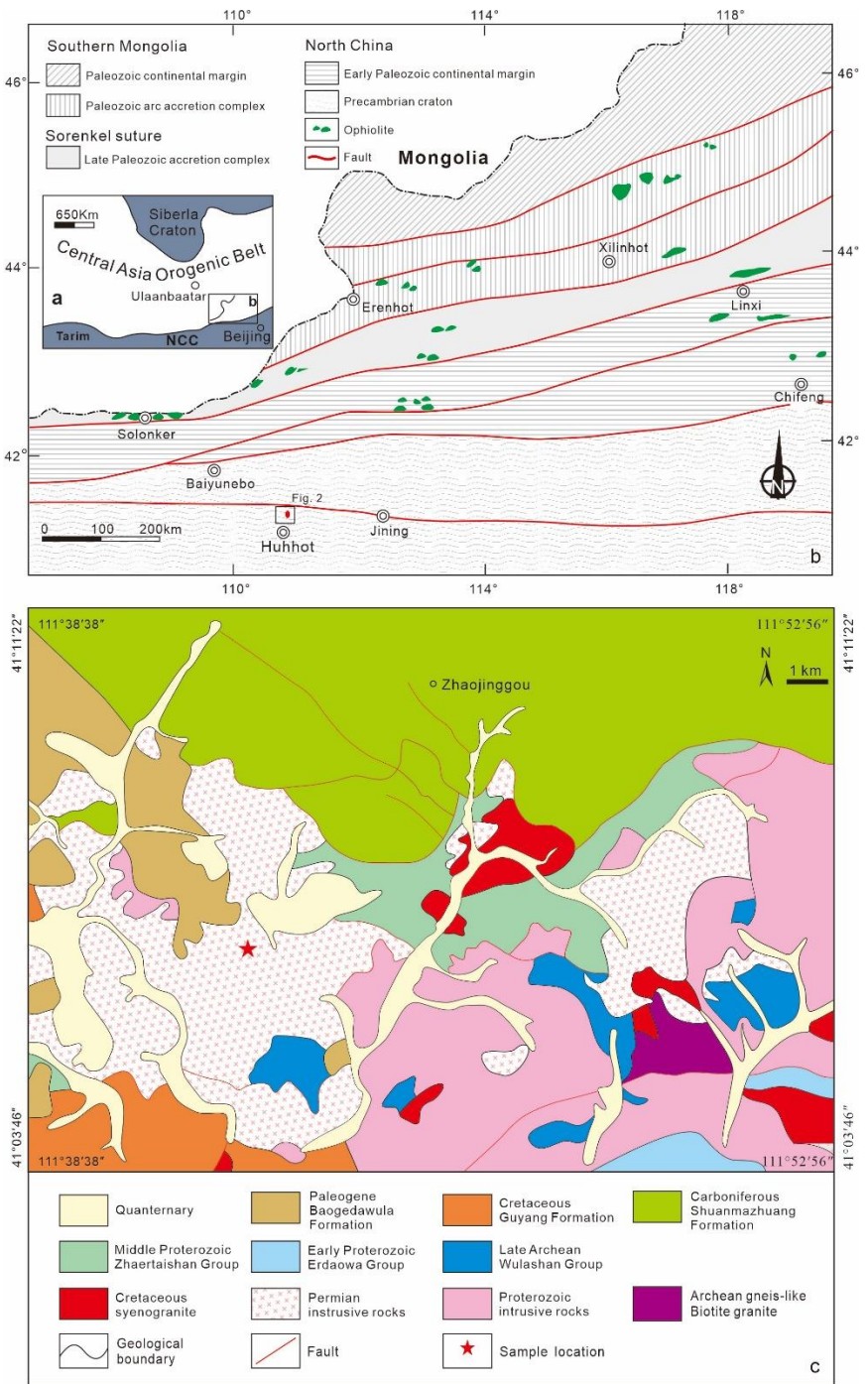

**Figure 1.** (**a**) Inset showing the location of the Central Asian Orogenic Belt and adjacent continents; (**b**) tectonic sketch showing the suture zone and the study area in Inner Mongolia (modified from [1]); (**c**) geological map of Zhaojinggou pluton and adjacent area (modified from [9]).

In this study, we present the petrology, geochemistry, zircon U-Pb-Hf isotope, and whole-rock Sr-Nd isotopic data to determine the crystallization age, petrogenesis, and

tectonic setting for the Zhaojinggou pluton. The results provide constraints on the timing of the closure of the eastern Paleo-Asian Ocean.

## 2. Geological Setting

### 2.1. The Paleo-Asian Ocean and The Central Asian Orogenic Belt (CAOB)

The Paleo-Asian Ocean, located between the Siberia plate and the Tarim-Alashan-North China plate, originated from the cracking of the Neoproterozoic Rodinia supercontinent. Accompanying the reduction and closure of the Paleo-Asian Ocean, the Tarim-Alashan-North China plate and Siberian plate collided and joined together, forming the Central Asian Orogenic Belt [1–3]. The CAOB, which is the Paleo-Asia Ocean tectonic domain, is the largest accretionary orogenic belt in the world [1]. From the Paleozoic to early Mesozoic, this orogenic belt experienced a complex tectonic evolution during the closure of the Paleo-Asian Ocean. It has long been an important window for studying the evolution history of the NCC and the Siberian plate, and it has received extensive attention (e.g., [6–9]).

The northern margin of the NCC is part of the CAOB [10–14]. A large-scale late Paleozoic to early Mesozoic magmatic massif developed in the northern margin of the NCC, sandwiched between the Siberian plate and the NCC. The magmatic massif was a significant record for the subduction and closure of the Paleo-Asian Ocean.

### 2.2. Regional Geology

The Zhaojinggou pluton is located in the eastern part of Wulashan-Daqingshan in the middle part of the northern NCC and the southern Linhe–Jining fault zone (Figure 1b). The geological evolution events in the area mainly included the Precambrian basement formation, the Mesoproterozoic rifting, the Late Paleozoic–early Mesozoic active continental margin evolution, and the Meso-Cenozoic rifting and uplifting process [15]. The early Precambrian strata are widely developed (Figure 1c; [9]). The late Archean Wulashan Group and the early Proterozoic Erdaowa Group constitute the crystalline basement. The Wulashan Group is mainly composed of amphibolite, amphibolite plagioclase gneiss, magnetite quartzite, granulite, and marble. The Erdaowa Group is a set of metamorphic basic volcanic–sedimentary rocks. The Zhaertaishan Group with low greenschist facies metamorphism formed in the middle Proterozoic rifting environment is widely distributed in the region, and its lithology is phyllite, slate, metamorphic sandstone, and marble. The Upper Carboniferous Shuanmazhuang Formation is widely distributed to the north and is composed of metaconglomerate and variegated metasandstone [16]. The Cretaceous Guyang Formation is exposed locally. The rock association is composed of yellow green conglomerate intercalated with gravelly lithic arkose, medium-grained arkose, coarse-grained lithic sandstone, siltstone, and black carbonaceous shale intercalated with seam.

Paleoproterozoic magmatism was extensive in the region. The Paleoproterozoic granitoid massif is composed of quartz diorite-diorite-hornblende monzonite and developed in the south of the Guyang–Wuchuan fault, Daqingshan Mountain, Inner Mongolia, and the western North China Craton yielded a zircon U-Pb age of 2416–2435 Ma [17]. Phanerozoic granites included Paleozoic biotite granite and granodiorite, such as Xiaojinggou biotite monzogranite that yielded a U-Pb aged 275 ± 1 Ma [18]. Mesozoic syenogranite, porphyry granite, and alkali feldspar granite complex were also mapped in the study area.

### 2.3. The Zhaojinggou Pluton

The Zhaojinggou pluton is located 2 km east of Wuchuan County, Hohhot city. The exposed area of the Zhaojinggou pluton is about 30 km$^2$, and the pluton mainly consists of monzogranite, without obvious lithofacies zoning. The medium-fine grained monzogranite is composed of quartz (25%), plagioclase (35%), potassium feldspar (35%), and scaly biotite (5%). The quartz, plagioclase, and K-feldspar show a xenomorphic granular texture, subhedral albite twins, and euhedral–subhedral plates, respectively.

## 3. Analytical Methods

### 3.1. Zircon U-Pb and Hf Isotopes

A representative sample (17ZG-1; 41°06′49.50″ N, 111°42′47.37″ E; Figure 1c) of Zhaojinggou monzogranite was collected for zircon U-Pb dating. The mineral separation was undertaken by Langfang Chengxin Geological Service Co., Ltd. The mounting and CL images of samples were completed in the laboratory of the Tianjin Center, China Geological Survey.

The fresh rock samples were mechanically crushed to 80 mesh, washed with water, magnetically separated, and then washed with alcohol. The single zircon mineral was selected manually; the zircon grains to be tested were made into targets with epoxy resin, and then ground to half of the zircon particles and polished. Zircon U-Pb age and Hf isotope analyses were performed in the Isotope Laboratory of the Tianjin Center, China Geological Survey, using a Neptune (LA-MC-ICPMS) manufactured by Thermo Fisher Scientific, Waltham, MA, USA. The spot diameter was 50 μm. The denudation time was 30 s. Zircon GJ-1 was used as the external standard to calculate the Hf isotope ratios, with a weighted mean $^{176}Hf/^{177}Hf$ ratio of $0.282012 \pm 35$ ($2\sigma$, n = 52) during our routine analyses. The detailed analytical and calculating procedure followed that of [19,20].

### 3.2. Whole-Rock Geochemistry

Six fresh rock samples without alteration were collected in the field (17ZG-2-1, 17ZG-2-2, 17ZG-2-3, 17ZG-2-4, 17ZG-2-5, and 17ZG-2-6). Whole-rock major oxides were measured using a PW4400/40 X-ray fluorescence spectrometer (XRF) at the laboratory of the Tianjin Center, China Geological Survey. The detailed analytical procedure is available in [21].

### 3.3. Rb-Sr and Sm-Nd Isotopic Compositions

Whole rock sample powders of about 200 mesh were dissolved by $HF + HClO_4 + HNO_3$ and reacted in a closed Teflon sample dissolver at a high temperature for 7 days. Total rare earth was obtained by separating the Rb and Sr from the dissolved sample with AG50W × 12 strong acid cation exchange resin, and then Sm and Nd elements were separated by HEHEHP resin (P507). The blank background of the whole process was stable at $Sm = 3.0 \times 10^{-11}$ g; $Nd = 5.4 \times 10^{-11}$ g. An Nd isotope ratio test was completed in Triton. The NIST SRM 987 Sr standard and the Jndi-1 Nd standard analyzed together with the international standard rock sample BCR-2 yielded $^{87}Sr/^{86}Sr = 0.705019 \pm 0.000005$ (2SE) and $^{143}Nd/^{144}Nd = 0.512636 \pm 0.000003$ (2SE), respectively. The detailed analytical procedure is available in [22–24].

## 4. Analytical Results

### 4.1. Zircon U-Pb Age

The zircon U-Pb results are shown in Table 1.

The zircon was columnar or ellipsoidal, colorless, and transparent, about 100 μm long and 50 μm wide. The cathodoluminescence image showed oscillatory zoning, and the Th/U ratios were greater than 0.1 (0.58–2.31), indicating magmatic origin. The $^{206}Pb/^{238}U$ age ranged from 283 to 292 Ma, and the weighted average age was $286.7 \pm 1.2$ Ma (*n* = 23; MSWD = 0.87) (Figure 2), representing the crystallization age of Early Permian Zhaojinggou monzogranite.

**Table 1.** Zircon U-Pb dating data from the Zhaojinggou monzogranite, Inner Mongolia.

| Sample Number | Content ($\times 10^{-6}$) | | Th/U | Isotope Ratio | | | | | | Age (Ma) | | | | | |
|---|---|---|---|---|---|---|---|---|---|---|---|---|---|---|---|
| 17ZG.1 Pb | U | Th | | $^{206}Pb/^{238}U$ | $1\sigma$ | $^{207}Pb/^{235}U$ | $1\sigma$ | $^{207}Pb/^{206}Pb$ | $1\sigma$ | $^{206}Pb/^{238}U$ | $1\sigma$ | $^{207}Pb/^{235}U$ | $1\sigma$ | $^{207}Pb/^{206}Pb$ | $1\sigma$ |
| Sam.1 26 | 506 | 539 | 1.0658 | 0.0449 | 0.0005 | 0.3282 | 0.0049 | 0.0530 | 0.0007 | 283 | 3 | 288 | 4 | 329 | 29 |
| Sam.2 20 | 411 | 240 | 0.5837 | 0.0457 | 0.0005 | 0.3301 | 0.0054 | 0.0524 | 0.0008 | 288 | 3 | 290 | 5 | 305 | 33 |
| Sam.3 14 | 257 | 276 | 1.0771 | 0.0457 | 0.0005 | 0.3305 | 0.0062 | 0.0525 | 0.0009 | 288 | 3 | 290 | 5 | 307 | 41 |
| Sam.4 29 | 502 | 1162 | 2.3144 | 0.0463 | 0.0005 | 0.3288 | 0.0075 | 0.0515 | 0.0012 | 292 | 3 | 289 | 7 | 264 | 51 |
| Sam.5 31 | 620 | 482 | 0.7781 | 0.0458 | 0.0005 | 0.3336 | 0.0053 | 0.0529 | 0.0007 | 288 | 3 | 292 | 5 | 323 | 31 |
| Sam.6 26 | 515 | 527 | 1.0216 | 0.0455 | 0.0005 | 0.3307 | 0.0079 | 0.0527 | 0.0013 | 287 | 3 | 290 | 7 | 315 | 57 |
| Sam.7 16 | 318 | 345 | 1.0847 | 0.0451 | 0.0005 | 0.3306 | 0.0059 | 0.0531 | 0.0009 | 285 | 3 | 290 | 5 | 335 | 36 |
| Sam.8 24 | 465 | 517 | 1.1114 | 0.0459 | 0.0005 | 0.3316 | 0.0055 | 0.0524 | 0.0008 | 289 | 3 | 291 | 5 | 301 | 33 |
| Sam.9 35 | 711 | 646 | 0.9091 | 0.0449 | 0.0005 | 0.3292 | 0.0046 | 0.0532 | 0.0006 | 283 | 3 | 289 | 4 | 338 | 28 |
| Sam.10 12 | 251 | 180 | 0.7171 | 0.0455 | 0.0005 | 0.3306 | 0.0069 | 0.0527 | 0.0010 | 287 | 3 | 290 | 6 | 317 | 43 |
| Sam.11 22 | 370 | 531 | 1.4368 | 0.0463 | 0.0005 | 0.3295 | 0.0071 | 0.0517 | 0.0009 | 292 | 3 | 289 | 6 | 271 | 42 |
| Sam.12 24 | 476 | 421 | 0.8845 | 0.0451 | 0.0006 | 0.3339 | 0.0077 | 0.0537 | 0.0010 | 285 | 3 | 293 | 7 | 357 | 42 |
| Sam.13 16 | 300 | 378 | 1.2604 | 0.0451 | 0.0005 | 0.3310 | 0.0060 | 0.0532 | 0.0009 | 285 | 3 | 290 | 5 | 337 | 36 |
| Sam.14 73 | 1163 | 2323 | 1.9973 | 0.0456 | 0.0005 | 0.3307 | 0.0044 | 0.0526 | 0.0006 | 287 | 3 | 290 | 4 | 313 | 27 |
| Sam.15 14 | 254 | 310 | 1.2183 | 0.0458 | 0.0005 | 0.3310 | 0.0077 | 0.0525 | 0.0011 | 288 | 3 | 290 | 7 | 306 | 48 |
| Sam.16 11 | 229 | 165 | 0.7212 | 0.0453 | 0.0005 | 0.3326 | 0.0076 | 0.0532 | 0.0011 | 286 | 3 | 292 | 7 | 337 | 48 |
| Sam.17 11 | 200 | 221 | 1.1072 | 0.0460 | 0.0005 | 0.3308 | 0.0070 | 0.0522 | 0.0010 | 290 | 3 | 290 | 6 | 293 | 45 |
| Sam.18 20 | 348 | 452 | 1.3011 | 0.0464 | 0.0005 | 0.3306 | 0.0057 | 0.0517 | 0.0007 | 292 | 3 | 290 | 5 | 272 | 33 |
| Sam.19 21 | 426 | 342 | 0.8014 | 0.0449 | 0.0005 | 0.3251 | 0.0055 | 0.0525 | 0.0008 | 283 | 3 | 286 | 5 | 305 | 35 |
| Sam.20 15 | 294 | 257 | 0.8739 | 0.0450 | 0.0005 | 0.3270 | 0.0074 | 0.0527 | 0.0011 | 284 | 3 | 287 | 6 | 317 | 48 |
| Sam.21 12 | 220 | 219 | 0.9940 | 0.0451 | 0.0005 | 0.3303 | 0.0062 | 0.0532 | 0.0009 | 284 | 3 | 290 | 5 | 336 | 39 |
| Sam.22 13 | 257 | 180 | 0.6987 | 0.0455 | 0.0005 | 0.3302 | 0.0064 | 0.0527 | 0.0009 | 287 | 3 | 290 | 6 | 315 | 41 |
| Sam.23 12 | 246 | 208 | 0.8456 | 0.0450 | 0.0005 | 0.3304 | 0.0072 | 0.0532 | 0.0011 | 284 | 3 | 290 | 6 | 337 | 45 |

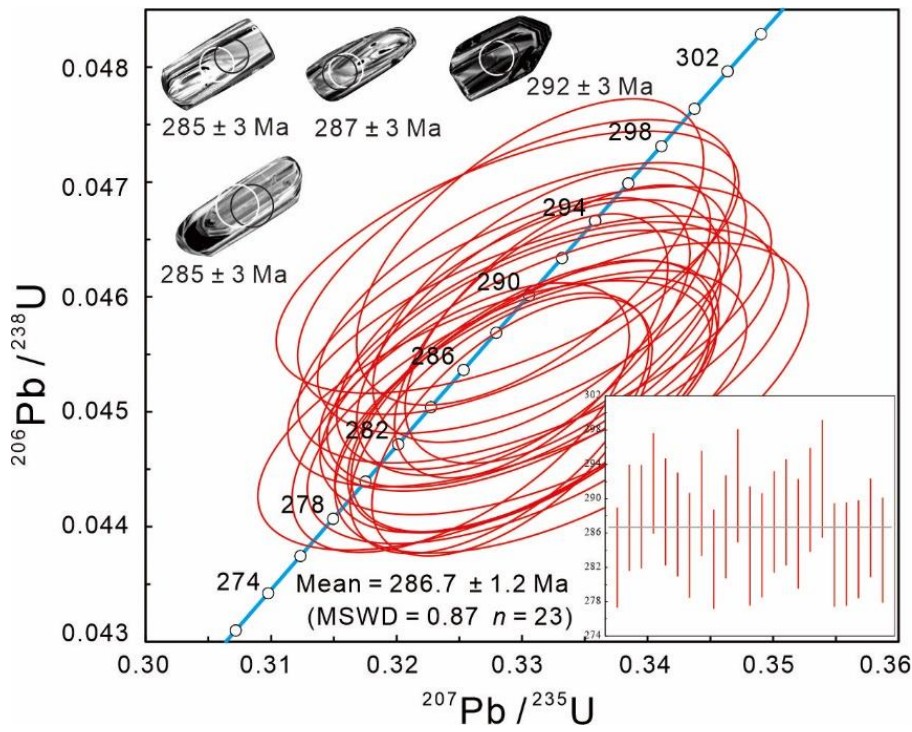

**Figure 2.** Zircon U-Pb concordia diagram for the sample from the Zhaojinggou monzogranite, Inner Mongolia.

### 4.2. Whole-Rock Geochemistry

The major and trace elemental compositions of six monzogranite samples from the Zhaojinggou pluton are listed in Table 2.

The samples were all in the granite area (Figure 3a).

**Table 2.** Major (wt. %) and trace element (ppm) data from the Zhaojinggou monzogranite, Inner Mongolia.

| Sample No. | 17ZG-2-1 | 17ZG-2-2 | 17ZG-2-3 | 17ZG-2-4 | 17ZG-2-5 | 17ZG-2-6 | Sample No. | 17ZG-2-1 | 17ZG-2-2 | 17ZG-2-3 | 17ZG-2-4 | 17ZG-2-5 | 17ZG-2-6 |
|---|---|---|---|---|---|---|---|---|---|---|---|---|---|
| $SiO_2$ | 73.32 | 75.22 | 74.44 | 74.21 | 72.92 | 74.60 | Sc | 0.91 | 0.84 | 0.96 | 1.01 | 1.42 | 1.01 |
| $Al_2O_3$ | 14.59 | 13.92 | 14.09 | 14.38 | 14.85 | 14.08 | Nb | 6.49 | 5.75 | 5.13 | 6.97 | 6.06 | 6.51 |
| $Fe_2O_3$ | 1.28 | 0.70 | 1.04 | 1.19 | 1.12 | 1.37 | Ta | 0.57 | 0.47 | 0.43 | 0.58 | 0.54 | 0.57 |
| FeO | 0.22 | 0.39 | 0.42 | 0.23 | 0.54 | 0.30 | Zr | 92.40 | 84.90 | 92.20 | 89.90 | 91.20 | 91.60 |
| CaO | 0.59 | 0.27 | 0.36 | 0.36 | 0.38 | 0.32 | Hf | 3.28 | 2.77 | 2.92 | 3.20 | 3.08 | 3.13 |
| MgO | 0.35 | 0.21 | 0.35 | 0.31 | 0.56 | 0.30 | Be | 1.59 | 1.47 | 1.37 | 1.85 | 1.46 | 1.84 |
| $K_2O$ | 3.51 | 2.89 | 3.73 | 3.35 | 3.88 | 3.31 | Ga | 13.30 | 12.20 | 11.80 | 12.90 | 15.60 | 12.50 |
| $Na_2O$ | 4.40 | 5.09 | 4.25 | 4.46 | 4.24 | 4.29 | Ge | 0.62 | 0.54 | 0.63 | 0.56 | 0.80 | 0.77 |
| $TiO_2$ | 0.22 | 0.21 | 0.21 | 0.21 | 0.21 | 0.21 | U | 1.52 | 0.96 | 0.64 | 0.65 | 0.52 | 0.70 |
| $P_2O_5$ | 0.07 | 0.06 | 0.06 | 0.06 | 0.07 | 0.06 | Th | 7.90 | 3.92 | 5.02 | 6.64 | 6.26 | 6.40 |
| MnO | 0.05 | 0.05 | 0.03 | 0.05 | 0.03 | 0.03 | F | 294.00 | 232.00 | 317.00 | 292.00 | 366.00 | 315.00 |
| LOI | 1.39 | 0.94 | 0.96 | 1.16 | 1.14 | 1.11 | La | 18.30 | 12.80 | 13.50 | 10.90 | 17.20 | 17.90 |
| Total | 99.99 | 99.96 | 99.95 | 99.97 | 99.94 | 99.98 | Ce | 28.60 | 16.30 | 27.80 | 27.50 | 28.40 | 29.30 |
| $Fe_2O_3^T$ | 1.52 | 1.13 | 1.51 | 1.45 | 1.72 | 1.70 | Pr | 3.32 | 2.46 | 2.39 | 1.98 | 3.18 | 3.11 |
| $FeO^T$ | 1.37 | 1.02 | 1.36 | 1.30 | 1.55 | 1.53 | Nd | 11.20 | 8.34 | 7.97 | 6.72 | 10.60 | 10.40 |
| $Mg^\#$ | 31.26 | 26.85 | 31.51 | 29.82 | 39.21 | 25.87 | Sm | 1.68 | 1.27 | 1.14 | 1.01 | 1.64 | 1.57 |
| $Na_2O/K_2O$ | 1.25 | 1.76 | 1.14 | 1.33 | 1.09 | 1.30 | Eu | 0.64 | 0.49 | 0.55 | 0.46 | 0.66 | 0.52 |
| A/CNK | 1.20 | 1.16 | 1.20 | 1.24 | 1.25 | 1.25 | Gd | 1.45 | 1.06 | 1.04 | 0.98 | 1.42 | 1.30 |
| A/NK | 1.32 | 1.21 | 1.28 | 1.31 | 1.33 | 1.32 | Tb | 0.19 | 0.14 | 0.12 | 0.13 | 0.18 | 0.16 |
| Cu | 3.28 | 7.60 | 6.55 | 7.43 | 2.94 | 6.60 | Dy | 0.88 | 0.62 | 0.58 | 0.59 | 0.92 | 0.70 |
| Pb | 14.80 | 29.60 | 9.13 | 6.74 | 7.95 | 6.26 | Ho | 0.16 | 0.11 | 0.10 | 0.11 | 0.18 | 0.13 |
| Zn | 24.60 | 13.20 | 23.40 | 17.70 | 30.60 | 18.60 | Er | 0.48 | 0.28 | 0.28 | 0.32 | 0.47 | 0.35 |
| Cr | 7.21 | 6.97 | 7.34 | 5.85 | 7.10 | 6.25 | Tm | 0.07 | 0.04 | 0.04 | 0.05 | 0.07 | 0.06 |
| Ni | 3.51 | 7.29 | 7.18 | 6.64 | 5.26 | 6.79 | Yb | 0.50 | 0.27 | 0.31 | 0.37 | 0.52 | 0.37 |
| Co | 2.42 | 2.14 | 2.23 | 2.39 | 2.48 | 2.83 | Lu | 0.08 | 0.05 | 0.05 | 0.06 | 0.08 | 0.06 |
| Li | 2.46 | 7.51 | 2.90 | 5.18 | 7.62 | 6.68 | Y | 4.49 | 2.54 | 2.58 | 2.69 | 4.25 | 3.47 |
| Rb | 86.80 | 54.90 | 90.00 | 81.20 | 101.00 | 80.00 | $\sum REE$ | 72.04 | 46.77 | 58.45 | 53.87 | 69.77 | 69.40 |
| Cs | 2.32 | 2.13 | 4.67 | 2.81 | 6.56 | 2.01 | δEu | 5.83 | 4.34 | 4.07 | 3.72 | 5.71 | 5.34 |
| Sr | 238.00 | 137.00 | 290.00 | 201.00 | 259.00 | 186.00 | $(La/Yb)_N$ | 24.08 | 31.39 | 28.79 | 19.33 | 21.77 | 31.81 |
| Ba | 1070.00 | 905.00 | 1040.00 | 903.00 | 1100.00 | 830.00 | $(La/Sm)_N$ | 6.63 | 6.14 | 7.20 | 6.56 | 6.38 | 6.94 |
| V | 17.50 | 15.90 | 14.40 | 19.00 | 19.80 | 15.70 | $(Gd/Yb)_N$ | 2.32 | 3.16 | 2.69 | 2.11 | 2.19 | 2.81 |

Note: All analyzed at Tianjin Institute of Geology and Mineral Resources, Tianjin, China. $Mg^\# = 100* Mg^{2+}/(Mg^{2+}+FeO^T)$; $FeO^T = FeO + 0.8998* Fe_2O_3$; $ACNK = Al_2O_3/(CaO + Na_2O + K_2O)$; $\delta Eu = Eu_N/(Sm_N \times Gd_N)^{1/2}$; N = chondrite-normalized; Primitive mantle values are available in [25].

The contents of $SiO_2$ were high, ranging from 72.92 to 75.22%, the $Al_2O_3$ contents ranged from 13.92 to 14.85%, and the contents of $Na_2O$ were 4.24 to 5.09%. The contents of CaO were low, ranging from 0.27 to 0.59%, the $K_2O$ values ranged from 2.89 to 3.88%, with an average of 3.45%; the contents of MgO were between 0.21 and 0.56%, with low–moderate $Mg^\#$ values (25.87–39.21) and low $Fe_2O_3^T$ values (1.13–1.72); and the $Na_2O/K_2O$ ratio was 1.09–1.76, with an average of 1.31. The $SiO_2$-$K_2O$ diagram (Figure 3b) shows calc–alkaline and high-k calc alkaline characteristics. The A/CNK-A/NK diagram (A/CNK: 1.16 to 1.25) shows that the samples were weak peraluminous (Figure 3c).

The trace element spider diagram shows that large ion lithophile elements such as Rb, Ba, K, and Sr were enriched, while high field strength elements such as Nb, Ta, and Ti were depleted (Figure 4a), showing the geochemical properties of subduction zone rocks. The REE distribution curve of granodiorite is obviously right dipping (Figure 4b), the $\sum REE$ value ranged from 46.77 to 72.04 ppm, the $(La/Yb)_N$ value was 19.33–31.81, the LREE was enriched, and the δEu mean value was 1.32 (1.11–1.54), showing a weak Eu positive anomaly.

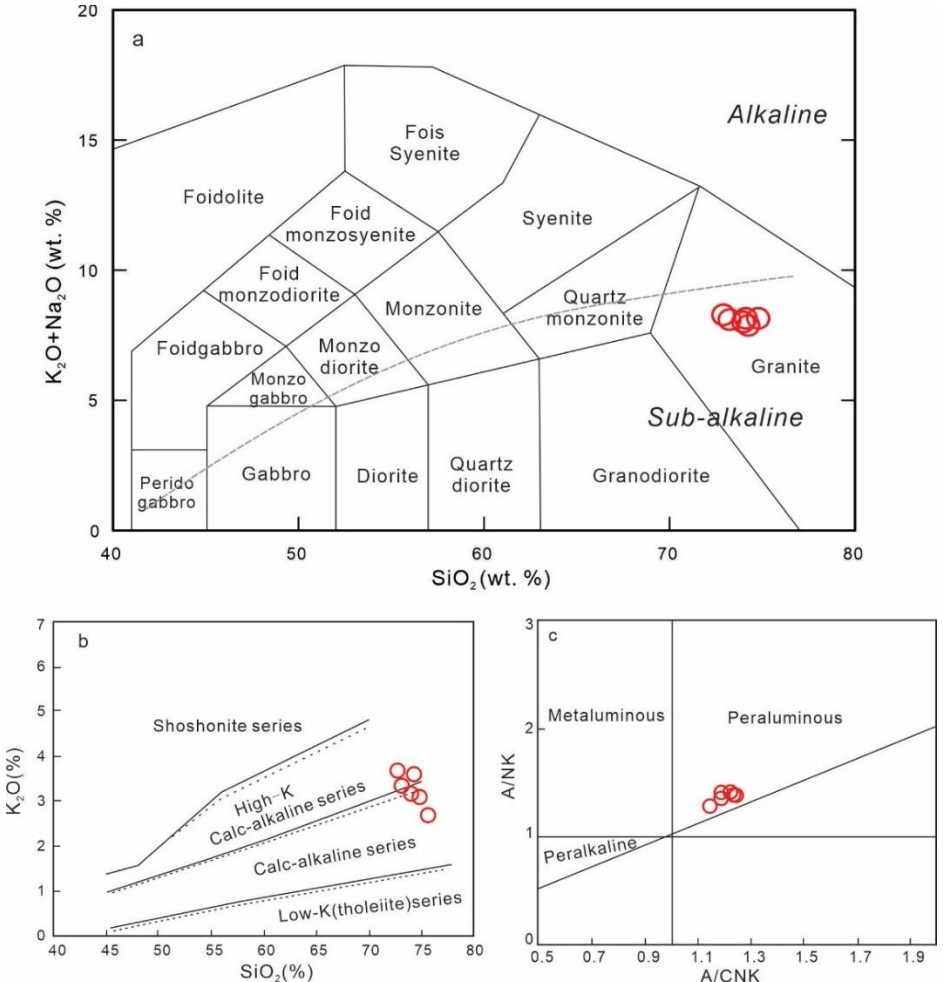

**Figure 3.** Classification of the samples from Zhaojinggou monzogranite pluton, Inner Mongolia. (**a**) TAS diagram [26]; (**b**) SiO$_2$-K$_2$O diagram of Zhaojinggou monzogranite, Inner Mongolia [26]; (**c**) A/CNK-A/NK diagram [27].

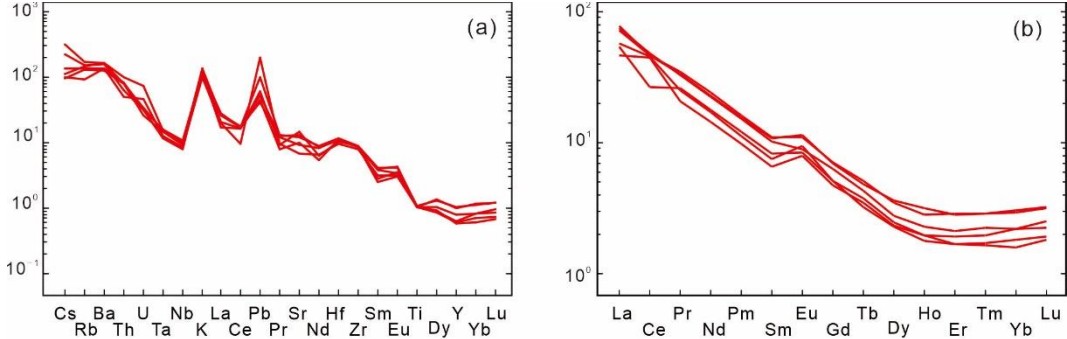

**Figure 4.** REE and trace elemental diagram of Zhaojinggou monzogranite in Inner Mongolia. (**a**) Primitive mantle-normalized trace elements spider diagram of Zhaojinggou monzogranites, Inner Mongolia. The values of normalization are from [28]. (**b**) Chondrite-normalized REE patterns diagram of Zhaojinggou monzogranite, Inner Mongolia (from [25]).

### 4.3. Zircon Hf Isotopes

Zircon in situ Hf isotopes compositions of monzogranite samples are listed in Table 3.

**Table 3.** Hf isotopic compositions for zircons of the Zhaojinggou monzogranite, Inner Mongolia.

| Spots | $t$(Ma) | $^{176}$Yb/$^{177}$Hf | 2σ | $^{176}$Lu/$^{177}$Hf | 2σ | $^{176}$Hf/$^{177}$Hf | 2σ | $\varepsilon_{Hf}$(0) | $\varepsilon_{Hf}$(t) | 2σ | $T_{DM1}$(Ma) | $T_{DM2}$(Ma) | $f_{Lu/Hf}$ |
|---|---|---|---|---|---|---|---|---|---|---|---|---|---|
| 1 | 283 | 0.061092 | 0.000936 | 0.002117 | 0.000007 | 0.282362 | 0.000027 | −14.51 | −8.69 | 0.96 | 1295 | 1846 | −0.94 |
| 2 | 288 | 0.032120 | 0.000213 | 0.001221 | 0.000011 | 0.282425 | 0.000023 | −12.26 | −6.18 | 0.84 | 1175 | 1692 | −0.96 |
| 3 | 288 | 0.051916 | 0.000446 | 0.001756 | 0.000011 | 0.282467 | 0.000033 | −10.79 | −4.80 | 1.16 | 1133 | 1605 | −0.95 |
| 4 | 292 | 0.030021 | 0.000349 | 0.001061 | 0.000009 | 0.282479 | 0.000024 | −10.36 | −4.15 | 0.87 | 1095 | 1567 | −0.97 |
| 5 | 288 | 0.024232 | 0.000140 | 0.000831 | 0.000005 | 0.282473 | 0.000028 | −10.57 | −4.39 | 1.00 | 1096 | 1580 | −0.97 |
| 6 | 287 | 0.075826 | 0.000652 | 0.002560 | 0.000012 | 0.282482 | 0.000024 | −10.25 | −4.43 | 0.87 | 1135 | 1581 | −0.92 |
| 7 | 285 | 0.054636 | 0.001581 | 0.001844 | 0.000033 | 0.282558 | 0.000026 | −7.57 | −1.66 | 0.94 | 1005 | 1405 | −0.94 |
| 8 | 289 | 0.047210 | 0.001065 | 0.001512 | 0.000026 | 0.282567 | 0.000027 | −7.23 | −1.16 | 0.95 | 982 | 1377 | −0.95 |
| 9 | 283 | 0.045350 | 0.000294 | 0.001772 | 0.000014 | 0.282370 | 0.000024 | −14.20 | −8.32 | 0.85 | 1271 | 1823 | −0.95 |
| 10 | 287 | 0.034523 | 0.000199 | 0.001163 | 0.000007 | 0.282507 | 0.000026 | −9.39 | −3.31 | 0.92 | 1059 | 1510 | −0.96 |
| 11 | 292 | 0.057211 | 0.000951 | 0.001874 | 0.000024 | 0.282470 | 0.000025 | −10.70 | −4.65 | 0.89 | 1132 | 1598 | −0.94 |
| 12 | 285 | 0.041712 | 0.001649 | 0.001415 | 0.000024 | 0.282614 | 0.000026 | −5.60 | 0.38 | 0.93 | 914 | 1275 | −0.96 |
| 13 | 285 | 0.050456 | 0.000167 | 0.001582 | 0.000008 | 0.282360 | 0.000024 | −14.58 | −8.63 | 0.85 | 1280 | 1844 | −0.95 |
| 14 | 287 | 0.052879 | 0.000320 | 0.001763 | 0.000009 | 0.282528 | 0.000024 | −8.64 | −2.66 | 0.87 | 1046 | 1470 | −0.95 |
| 15 | 288 | 0.052959 | 0.000521 | 0.002057 | 0.000027 | 0.282342 | 0.000030 | −15.21 | −9.27 | 1.06 | 1322 | 1887 | −0.94 |
| 16 | 286 | 0.032289 | 0.000677 | 0.001211 | 0.000015 | 0.282377 | 0.000023 | −13.95 | −7.90 | 0.83 | 1242 | 1799 | −0.96 |

Note: $\varepsilon_{Hf}(0) = [(^{176}Hf/^{177}Hf)_s/(^{176}Hf/^{177}Hf)_{CHUR,0}-1] \times 10{,}000$; $\varepsilon_{Hf}(t) = \{[(^{176}Hf/^{177}Hf)_s - (^{176}Lu/^{177}Hf)_s \times (e^{\lambda t} - 1)]/[(^{176}Hf/^{177}Hf)_{CHUR,0} - (^{176}Lu/^{177}Hf)_s \times (e^{\lambda t} - 1)] - 1\} \times 10{,}000$; $T_{DM1} = 1/\lambda \times \ln\{1 + [(^{176}Hf/^{177}Hf)_s - (^{176}Hf/^{177}Hf)_{DM}]/[(^{176}Lu/^{177}Hf)_s - (^{176}Lu/^{177}Hf)_{DM}]\}$; $T_{DM2} = T_{DM1} - (T_{DM1}$-t$) \times [(f_{cc} - f_s)/(f_{cc} - f_{DM})]$; $f_s = (^{176}Lu/^{177}Hf)_s/(^{176}Lu/^{177}Hf)_{CHUR}-1$, $(^{176}Lu/^{177}Hf)_s$ and $(^{176}Hf/^{177}Hf)_s$ are the measured value of the samples, $(^{176}Lu/^{177}Hf)_{CHUR} = 0.0332$, $(^{176}Hf/^{177}Hf)_{CHUR,0} = 0.282772$; $(^{176}Lu/^{177}Hf)_{DM} = 0.0384$, $(^{176}Hf/^{177}Hf)_{DM} = 0.28325$, $f_{cc}$, $f_s$, $f_{DM}$ represent the average $f_{Lu/Hf}$ from continental crust, samples and depleted mantle, $f_{cc} = -0.55$, $f_{DM} = 0.16$, $\lambda = 1.867 \times 10^{-11}$ a$^{-1}$, $t$ = Formation time of sample zircon.

The zircon in situ Hf isotopic results showed that the $^{176}$Hf/$^{177}$Hf ratios ranged from 0.282342 to 0.282614, and the $^{176}$Lu/$^{177}$Hf ratios ranged from 0.000831 to 0.002560 (Figure 5a,b). Zircon grains from the monzogranite samples yielded $\varepsilon_{Hf}$(t) values of −9.27–0.38 (mean −4.74), a $T_{DM1}$ of 914–982 Ma, and a $T_{DM2}$ of 1275–1887 Ma.

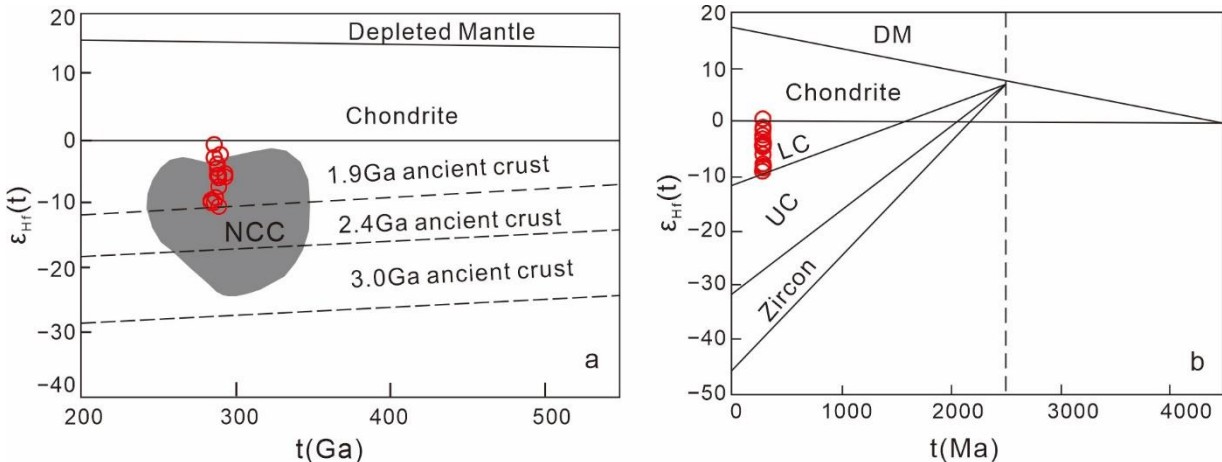

**Figure 5.** $\varepsilon_{Hf}$(t)-t diagrams of the Zhaojinggou monzogranite. (**a**) Hf isotope contrast with NCC; the gray field indicates Carboniferous–Permian intrusive rocks on the northern margin of the NCC (after [29]). (**b**) Hf isotope evolution.

*4.4. Whole-Rock Rb-Sr and Sm-Nd Isotopic Compositions*

The whole-rock Sr–Nd isotopic compositions for Zhaojinggou monzogranite samples are listed in Table 4 and illustrated in Figure 6.

The initial Sr-Nd isotopic compositions of granites were rectified by the corresponding zircon U-Pb age of 286.7 Ma. The $^{87}$Sr/$^{86}$Sr values of granites ranged from 0.715233 to 0.716612, and the average value was 0.7160, which was obviously higher than the original mantle value ($^{87}$Sr/$^{86}$Sr = 0.7045, [23]). The $^{143}$Nd/$^{144}$Nd values of granites ranged from 0.511989 to 0.512021, and the average value was 0.512000, which was slightly lower than the original mantle value ($^{143}$Nd/$^{144}$Nd = 0.512638, [24]). On the basis of the zircon U-Pb ages, the granite samples had ($^{87}$Sr/$^{86}$Sr)$_i$ values of 0.712345–0.713723, an average value

of 0.7128; $\varepsilon_{Nd}(t)$ values of −8.89−−8.21, an average value of −8.56; and the $T_{DM2}$ was 1718–1773 Ma.

**Table 4.** Whole-rock Sr-Nd isotopic composition of the Zhaojinggou monzogranite, Inner Mongolia.

| Sampe No. | Rock | Age(Ma) | $^{87}Rb/^{86}Sr$ | $^{87}Sr/^{86}Sr$ | $2\sigma$ | $(^{87}Sr/^{86}Sr)_i$ | $^{147}Sm/^{144}Nd$ | $^{143}Nd/^{144}Nd$ | $2\sigma$ | $\varepsilon_{Nd}(t)$ | $T_{DM}(Ma)$ | $T_{DM2}(Ma)$ |
|---|---|---|---|---|---|---|---|---|---|---|---|---|
| 17ZG-2-1 | Monzogranite | 286.7 | 0.708098 | 0.716612 | 0.000009 | 0.713723 | 0.090670 | 0.511996 | 0.000007 | −8.65 | 1427.5 | 1754 |
| 17ZG-2-2 | Monzogranite | 286.7 | 0.707833 | 0.715951 | 0.000007 | 0.713063 | 0.092048 | 0.512021 | 0.000008 | −8.21 | 1412.5 | 1718.4 |
| 17ZG-2-3 | Monzogranite | 286.7 | 0.707661 | 0.715614 | 0.000009 | 0.712727 | 0.086461 | 0.512001 | 0.000010 | −8.40 | 1374.6 | 1733.8 |
| 17ZG-2-4 | Monzogranite | 286.7 | 0.708033 | 0.715233 | 0.000007 | 0.712345 | 0.090850 | 0.511997 | 0.000008 | −8.63 | 1428.4 | 1752.9 |
| 17ZG-2-5 | Monzogranite | 286.7 | 0.707985 | 0.715373 | 0.000008 | 0.712485 | 0.093521 | 0.511989 | 0.000003 | −8.89 | 1470.1 | 1773.4 |
| 17ZG-2-6 | Monzogranite | 286.7 | 0.708507 | 0.715373 | 0.000009 | 0.712483 | 0.091251 | 0.512000 | 0.000004 | −8.59 | 1429.3 | 1749.3 |

Note: $(^{87}Rb/^{86}Sr)_{CHUR}$ = 0.0847; $(^{87}Sr/^{86}Sr)_{CHUR}$ = 0.7045; $(^{147}Sm/^{144}Nd)_{CHUR}$ = 0.1967; $(^{143}Nd/^{144}Nd)_{CHUR}$ = 0.512638 ([30]); *t*-Diagenetic age.

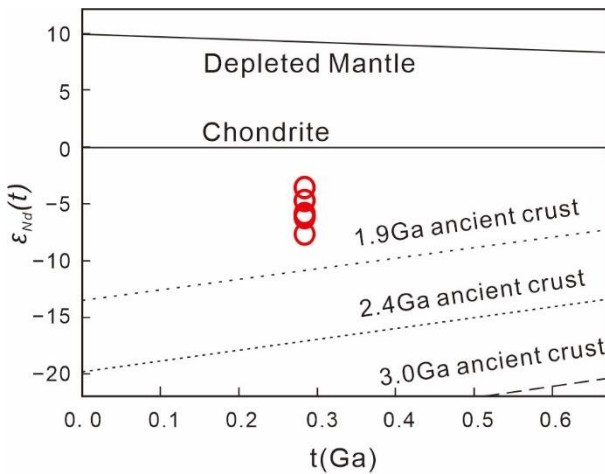

**Figure 6.** $\varepsilon_{Nd}(t)$-t diagrams of the Zhaojinggou monzogranite.

## 5. Discussion

### 5.1. Timing of the Magmatism on Northern NCC in Early Permian

Magma is the carrier of material and energy exchange in plate activity; its composition, distribution, and evolution have a close relationship with the tectonic environment, and it is often used to reconstruct the paleotectonic environment [31]. The southward subduction of the Paleo-Asian oceanic crust and subsequent collision caused a large number of magmatic records in the northern margin of the NCC [11]. Large-scale magmatic plutons occurred in the Early Permian in the northern NCC, stretching for hundreds of kilometers from west to east, including the Wulatzhongqi biotite monzogranite (279 ± 3 Ma, [15]); the Bayan Obo granodiorite and biotite granite (269 ± 3 Ma, 268 ± 2 Ma, [32]); the Siziwangqi syenogranite and biotite granodiorite (264 ± 3 Ma, 267 ± 9 Ma, [33]); and the granodiorite and diorite in the Chengde area (288 ± 2 Ma, [34]). These ages prove extensive late Permian magmatic events in the northern NCC, indicating strong subduction and signs of oceanic ridge–trench collision [10].

In this study, we obtained the Late Permian crystallization age of the Zhaojinggou monzogranite pluton (286.7 ± 1.2 Ma), which is basically consistent with the above ages from the northern NCC. These magmatic records are an important basis for determining the closing time limit of the Paleo-Asian Ocean.

### 5.2. Petrogenesis of the Zhaojinggou Granite Pluton

On the AMF-CMF petrogenetic diagram, the samples are distributed in the zone of partial melts of metapelitic sources, near the left boundary of partial melts from meta-greywackes (Figure 7a), indicating that sedimentary rocks rich in aluminum may be the source [35].

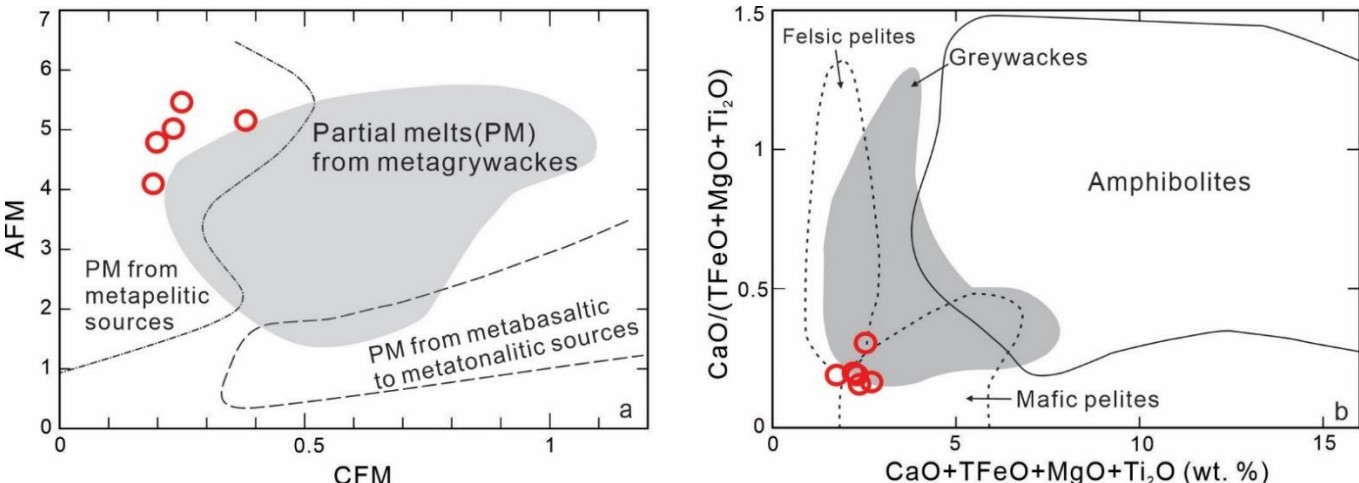

**Figure 7.** Diagrams for the petrogenesis of the Zhaojinggou monzogranite. (**a**) AMF (molar $Al_2O_3/(MgO + TFeO)$) versus CMF (molar $CaO/(MgO + TFeO)$) diagram (from [25]); (**b**) $CaO/(TFeO + MgO + TiO_2)$ versus $CaO+TFeO+MgO+TiO_2$ diagram (modified from [30]).

Furthermore, all of the samples were plotted within or near the field of mafic pelites or greywackes (Figure 7b), indicating that the source rocks experienced intracrustal recycling [36]. Additionally, the samples with the middling fractionated REE patterns and smooth HREE patterns may have originated from mid-crustal materials containing plagioclase or plagioclase + biotite. The obviously geochemical characteristics of high-K calc-alkaline, A/CNK>1, enrichment in LREE and LILE, and depletion in HFSE may reveal the features of Andean-type continental arc magmas. Combined with the low Sr isotope content (<300 ppm) and sedimentary origin, it can be further inferred that the pluton is S-type granite, which is usually produced in a subduction environment [37].

Moreover, the characteristics of the low–moderate $Mg^{\#}$ values (25.87–39.21), low $Fe_2O_3^T$ values (1.13–1.72), weak positive δEu average values of 1.32 (1.11–1.54), relatively high $(^{87}Sr/^{86}Sr)_i$ values of 0.712345–0.713723, low $\varepsilon_{Nd}(t)$ values of −8.89−−8.21 (average value of −8.56), a $T_{DM2}$ of 1718–1773 Ma (Figure 6), low $\varepsilon_{Hf}(t)$ values of −9.27–0.38 (mean −4.74), a $T_{DM2}$ of 1275–1887 Ma, and the points of $\varepsilon_{Hf}(t)$-t primitively falling within the range of the Carboniferous–Permian intrusive rocks on the northern margin of the NCC (Figure 5a) indicate that the pluton was derived from late Paleoproterozoic to Mesoproterozoic lower crustal mafic materials in the NCC.

### 5.3. Tectonic Implications

The tectonic evolution of the northern margin of the NCC during the Permian period has been a controversial issue [4]. Zhou et al. (2019) [38] hold that the tectonic system of the northern margin of the North China Craton changed from a subduction (pre-275 Ma) to post-collision extension (275–255 Ma). In contrast, the arc–trench system within the ocean played a leading role during the Permian [10]. The origin of the Zhaojinggou granite pluton provides new constraints on the tectonic evolution of the northern margin of NCC in the eastern margin of the Paleo-Asian Ocean. According to the zircon Hf isotopes, negative $\varepsilon_{Hf}(t)$ values, and various $T_{DM2}(Hf)$ ages (1275–1887 Ma), the metapelite–metagreywacke protolith of the Zhaojinggou pluton could have been formed due to Mesoproterozoic sources with suptasubduction affinity. The geochemical results suggest that the granite had Ni, Nb, and Ti negative abnormality, indicating the involvement of oceanic sediments. Therefore, we propose that the pluton was derived from partial melting of the lower crust in an active continental margin environment during the late Paleoproterozoic southward subduction of the Paleo-Asian Oceanic lithosphere. Additionally, Zhang et al. holds that the gabbroic diorite (291.1 ± 1.8 Ma) located in the Zhaojinggou area was derived from the subduction-related enriched subcontinental lithospheric mantle (SCLM) [39].

Wang et al. (2007) [34] also suggest that the northern margin of the NCC was an Andean active continental margin in the Early Permian by studying the diorite plutons (288 Ma, consistent with the age in this article) that intruded in the late Paleozoic. Furthermore, a classical model [40] can well explain the genesis of the pluton: dehydration of the subduction plate → uneven hydration of the mantle wedge → rising of the mantle-derived magma → crust melting → differentiation and crystallization of magmas (Figure 8a). Here, the Zhaojinggou granite pluton yielded a zircon U-Pb age of ~287 Ma and indicates Early Permian emplacement. Thus, we propose the Paleo-Asian Ocean did not shut until the Early Permian in the northern margin of the NCC (Figure 8b).

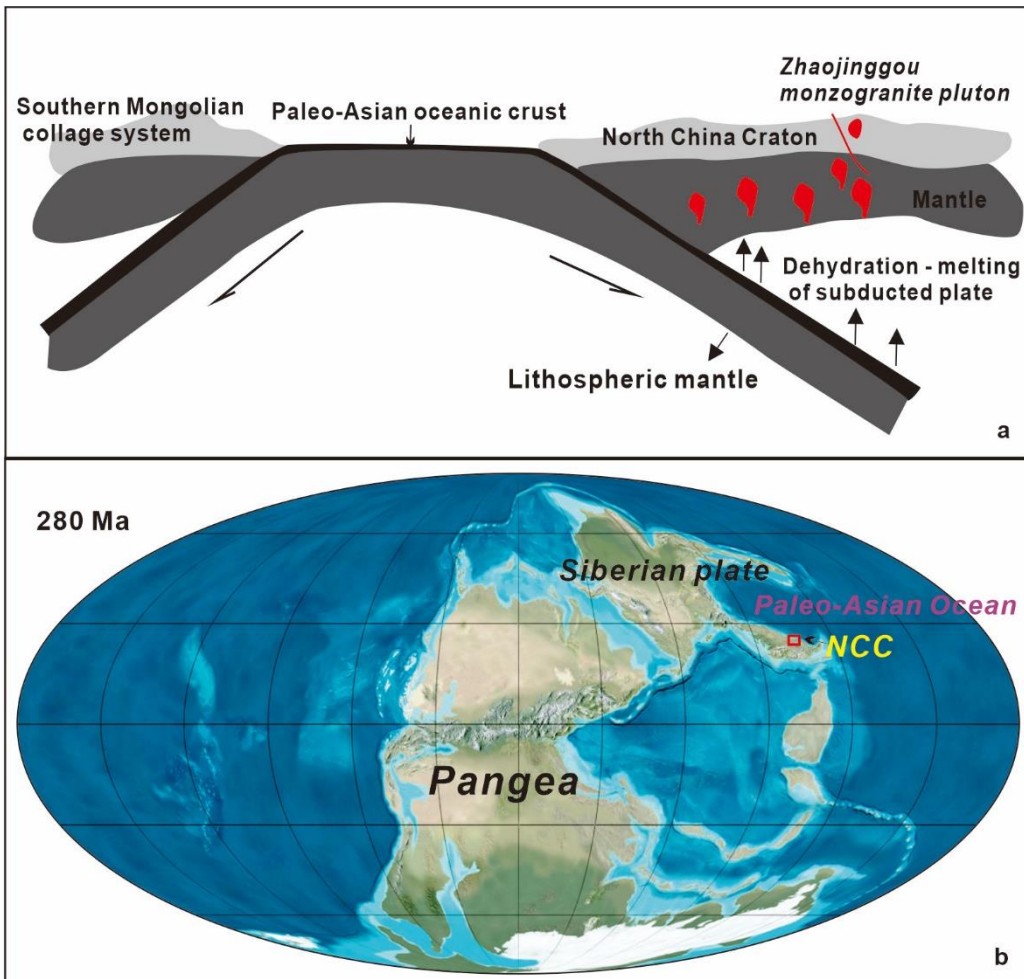

**Figure 8.** (**a**) Schematic diagram of the tectonic setting for the Zhaojinggou area in the northern NCC during the Early-Middle Permian (modified from [1,6,40]). (**b**) The Paleogeographic earth reconstruction, Early Permian (modified from [41]).

## 6. Conclusions

1. The Zhaojinggou pluton yielded a zircon U-Pb age of 286.7 ± 1.2 Ma, indicating an Early Permian magmatism.
2. The Zhaojinggou monzogranite was derived from lower crustal melting in a continental arc setting, suggesting the closure of the eastern Paleo-Asian Ocean in the north margin of the NCC postdated the Early Permian.

**Author Contributions:** Conceptualization: G.-Y.L., L.Q. and Z.-D.L.; methodology: L.G. and J.-R.T.; software: Q.Z. and T.-F.G.; validation: L.Q. and J.-Y.W.; investigation: G.-Y.L., Z.-D.L. and C.F.; data curation: G.-Y.L.; writing—original draft preparation: G.-Y.L., L.Q. and Z.-D.L.; writing—review and editing: G.-Y.L. and L.Q. All authors have read and agreed to the published version of the manuscript.

**Funding:** This research was funded by the China Geological Survey (Grant No.12120113057300; DD20190119; DD20211191; DD20221686) and National Natural Science Founding of China (Grant No.41502082).

**Institutional Review Board Statement:** Not applicable.

**Informed Consent Statement:** Not applicable.

**Data Availability Statement:** Not applicable.

**Conflicts of Interest:** The authors declare no conflict of interest.

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
