# Peer review of "Closure of the Eastern Paleo-Asian Ocean: Constraints from the Age and Geochemistry of Early Permian Zhaojinggou Pluton in Inner Mongolia (North China)"

_minerals, doi:10.3390/min12060738_

Round 1

Reviewer 1 Report

Dear Authors,

The manuscript is presented geochronological, isotopic, and geochemical studies of the Zhaojinggou massif at northern margin of the North China Craton. The significance of this study is that the age of the massif may mark the beginning of the closure of the Paleo-Asian Ocean.

The authors obtained new important data indicating that subduction was still going on at the northern margin of the NCC about 288 Ma. This result contributes to the active debate on the Paleo-Asian ocean closure time.

I think that as contribution to solving the Central Asian Orogenic Belt evolution problems the manuscript is publishable. However, in present form, it has many problems, see comments below. I think that publication of this manuscript is possible only after a fundamental revision. I recommend either a major revision or, if the authors do not have enough time to reworking the article, its re-submission.

I provide some comments below.

Major comments

Terminology is often incorrect. Authors either use terms incorrectly, such as “hypabyssal” for sedimentary rocks, or use synonyms not accepted in academic style instead of terms, such as “cobweb diagram” instead of “spider diagram”.

The method description is too compressed. It is necessary to give the methods in detail or refer to works with a detailed description of the methods. It is also necessary to give the results of standard measurement for isotope analysis.

Geochemical and isotopic data presentation is not very careful. There are no rock names of samples, number of significant figures is incorrect, some value (Eu anomaly) is calculated incorrectly and etc.

There is no comparison with the data of other authors who have studied the Zhaojinggou massif. The authors mention granodiorite and diorite in Chengde area (288 ± 2 Ma) with the same age as Zhaojinggou massif, but they do not make a comparison between these simultaneous magmatic events.

The Discussion section don’t present in-depth analysis of the data obtained by authors. Hf- Nd isotopic relationships are not discussed in any way, no Sr isotope ratios are looked at. The arguments in support of the suprasubduction nature of the Zhaojinggou massif are insufficient. The Nd isotopic data do not show the contribution of juvenile material. Some features (high Al, high 87Sr/86Sr ratios) indicate that the pluton  has S-type granite affinity. You need a more in depth discussion of the geochemistry and isotopy of the granites.

Minor comments

Line 43 “crustal” not “strastal”

Lines 70 and 71 the term “hypabyssal” commonly used for magmatic rocks, not for sediments

Lines 88 and 89. A petrography is not understood. Please check the use of terms such as flake for twins in plagioclase. Maybe you meant laminar twins? Or tabular habitus instead of “plate” for K fieldspar?

Lines 134-137 the text replicates the Table 3 very much

Line 139  remove “mass” from “shows that the rock mass is calc alkaline high-k calc alkaline”

Line 172 “cobweb diagram” is not correct, please, use term “spider diagram”

Line 177 and Table 3 what is “δEu”? One calculates Eu anomaly as Eu/Eu* = EuN/(SmN*GdN)1/2 and if you calculate the Eu anomaly this way, the value will be 1.1 – 1.5, not 3.7-5.8. Your figure 4b does not show such a huge anomaly, it shows an anomaly no more than 1.5.

Line 193. Remove “Supplementary” from “Whole-rock Sr–Nd isotopic compositions for Zhaojinggou monzonitic granite samples are listed in Supplementary Table 4”

Line 194 It is not clear whether Sr and Nd isotopic data are given for gabbroic diorites or granites. The values 0.712345–0.713723 can unlikely be considered as low

Lines 236-238. This suggestion is unclear. How was the massif formed by paritial melting in the Mesoproterozoic if it is Permian in age? It should be more clearly written that the metapelite-metagreywacke protolith of the Zhaojiagou pluton could have been formed due to Mesoproterozoic sources with suptasubduction affinity.

Line 239. Why do you say that the Zhaojiagou massif was formed in an arc condition? Your own isotopic data and Figure 8 indicate that it was originated under an active continental margin environment

Fig. 1. Please, ether adds to the discussion data on samples from “other works” and references these works, or removes this sign from the legend and from the map. It is recommended to combine the signs of the Permian granite, fine-grained granite and gabbroic diorite under the header of the Zhaojinggou massif. Description of Precambrian regional geology is not corresponds with map on fig, 1c: there are Groups on the map without description in the text. Where is the Xiaojinggou biotite monzogranite with the Permian age (275 Ma) on fig. 1c?

Fig. 5, Please, give reference for source of diagrams

Table 3 - number of significant figures should be improved. The number of significant figures in a measurement, such as 2.531, is equal to the number of digits that are known with some degree of confidence (2, 5, and 3) plus the last digit (1), which is an estimate or approximation (Rules for Using Significant Figures)

Table 4 Replace Chinese with English in the column headers. It is not clear whether Sr and Nd isotopic data are given for gabbroic diorites or granites. Please, give rock name for samples.

Author Response

Thank you very much for the constructive comments! The word document with comments and track changes are very helpful! We learnt a lot! Thanks!

Response for your comments lists in the attachment.

Reviewer 2 Report

Provide the information about Paleo-Asian Ocean, time of origin, rocks originated in this ocean and so on. Provide paleogeographic map.

Author Response

Thank you very much for the constructive comments! The word document with comments and track changes are very helpful! We learnt a lot! Thanks!

Response to your comments lists in the attachment.

Reviewer 3 Report

This work presents many geochemical and geochemical isotopic data from a few samples of granitic rocks. Despite the large amount of data, deductions about the origin of granites are hasty and not well demonstrated. The authors discuss only some of the data collected. No comments are made on the Rb-Sr and Sm-Nd isotopic data indicating a clear crustal origin of the melts. They also point out that some isotopic data were collected on samples that were not chemically analysed (row 193-194 Sr/Sr and Sm/Nd on gabbroic diorite, but in tablet 4-3 they are monzonitic granite). The diagrams of Fig. 5 are obsolete. The crustal origin of the granites is well documented by the collected data, while the tectonic environment related to subduction is not congruently inferred. A discussion on the more mafic rocks can provide further indications. In addition, an accurate mineral chemistry can reveal the origin of magmas  and the tectonic setting.

The collected data on zircon ages are valid.

In the introduction section, the aim of paper is not explained in detail. It must be explained why the age of pluton is related to the age of the ocean's closure.

Some corrections and suggestions can be found in the attached pdf file.

Author Response

Thank you very much for the constructive comments! The word document with comments and track changes are very helpful! We learnt a lot! Thanks!

Response to your comments lists in the the attachment.

Round 2

Reviewer 1 Report

Dear authors,

please, correct the number of significant figures in the table 3, if you give 3 significant figures, they should be 3 significant figures everywhere

Reviewer 3 Report

The manuscript is improved from the previous version, however, the English needs to be improved. I suggest an extensive revision of the English. Few corrections in the attached file

Author Response

This manuscript is a resubmission of an earlier submission. The following is a list of the peer review reports and author responses from that submission.